# 3D Meta-Registration: Meta-learning 3D Point Cloud Registration Functions

## Abstract

Learning robust 3D point cloud registration functions with deep neural networks has emerged as a powerful paradigm in recent years, offering promising performance in producing spatial geometric transformations for each pair of 3D point clouds. However, 3D point cloud registration functions are often generalized from extensive training over a large volume of data to learn the ability to predict the desired geometric transformation to register 3D point clouds. Generalizing across 3D point cloud registration functions requires robust learning of priors over the respective function space and enables consistent registration in presence of significant 3D structure variations. In this paper, we proposed to formalize the learning of a 3D point cloud registration function space as a meta-learning problem, aiming to predict a 3D registration model that can be quickly adapted to new point clouds with no or limited training data. Specifically, we define each task as the learning of the 3D registration function which takes points in 3D space as input and predicts the geometric transformation that aligns the source point cloud with the target one. Also, we introduce an auxiliary deep neural network named 3D registration meta-learner that is trained to predict the prior over the respective 3D registration function space. After training, the 3D registration meta-learner, which is trained with the distribution of 3D registration function space, is able to uniquely parameterize the 3D registration function with optimal initialization to rapidly adapt to new registration tasks. We tested our model on the synthesized dataset ModelNet and FlyingThings3D, as well as real-world dataset KITTI. Experimental results demonstrate that 3D Meta-Registration achieves superior performance over other previous techniques (e.g. FlowNet3D).

## 1 Introduction

The point cloud registration is defined as a process to determine the spatial geometric transformations (i.e. rigid and non-rigid transformation) that can optimally register the source point cloud towards the target one. In comparison to classical registration methods Besl & McKay (1992); Yang et al. (2015); Myronenko et al. (2007), learning-based registration methods Liu et al. (2019); Balakrishnan et al. (2018) usually leverage a neural network-based structure to directly predict the desired transformation for a given pair of source and target point clouds.

Recently based on the PointNet Qi et al. (2017a) structure, Liu et al. proposed FlowNet3D Liu et al. (2019) to learn the points flow field to register two point clouds together. Balakrishnan et al. proposed VoxelMorph Balakrishnan et al. (2018) for aligning two volumetric 3D shapes. These methods achieved impressive performance for the registration task of 3D shapes/scenes. In comparison to iterative registration methods, learning-based methods have advantages in dealing with a large number of datasets since learning-based methods can transfer the registration pattern from one dataset to another one. However, there are two main challenges for the learning-based methods. Firstly, learning-based networks often require a large volume of data and a long learning period to acquire the ability to predict the desired geometric transformation to register 3D point clouds. Secondly, The generalization capacity can be greatly degraded if the distribution of the dataset in practice differs from the training dataset.

As shown in Figure 1, previous learning-based methods tend to learn a single 3D registration function (learner) for any pair of source and target point clouds. In this way, the 3D registration learner

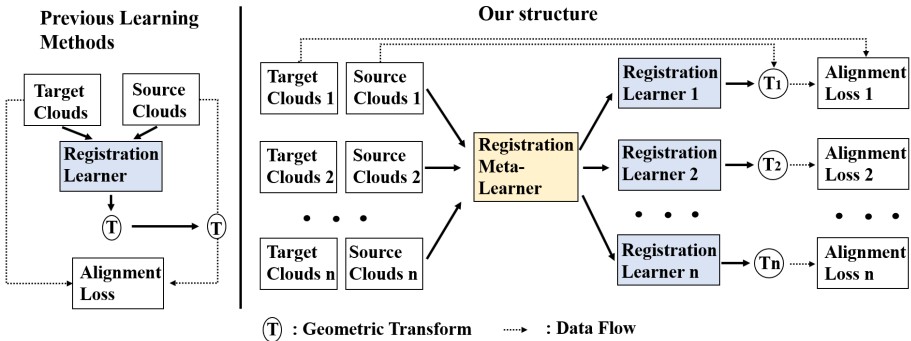

Figure 1: Comparison of the pipeline between previous learning methods and our solution for point cloud registration. Previous learning-based methods regard 3D registration as a task for the given point clouds. In comparison, we define each task as learning a unique 3D registration function (learner) for each pair of given 3D objects.

performs well from extensive training on a large number of labeled data. In contrast, we proposed to formalize the learning of a 3D registration function space as a meta-learning problem. More specifically, we define each task as learning a unique 3D registration function (learner) for each given 3D object. Besides, we design an auxiliary deep neural network as a meta-learner that can predict the prior over the respective 3D registration function space. The meta-learner is responsible for providing the optimal initialization of a 3D registration learner. In general, our meta-learning-based approach gains competitive advantages over existing generalization methods that our method can uniquely parameterize the 3D registration function for each pair of shapes to provide 3D point cloud registration.

As shown in Figure 2, our model includes two modules: 3D registration learner and 3D registration meta-learner. We observe that different regions of a 3D shape vary in their geometric structures which makes it more sense that we have a region-conditioned transformation instead of the shape-conditioned one. In this way, the proposed 3D registration learner includes shape embedding and region aware flow regression which uses multiple non-linear multi-layer perceptron (MLP)-based function to predict the transforms and weights for different regions respectively. The 3D registration meta-learner includes two parts. The first part is a variational auto-encoder that maps shape representations to a distribution over 3D registration tasks, which gives the priors over the 3D registration function space and the second part is to sample from the 3D registration function space to predict optimal initialization for the parameters of the 3D shape registration function. Our contributions are listed as below:

- In this paper, to the best of our knowledge, it is the first time to formalize learning of a 3D point cloud registration function space as a meta-learning problem 3D computer vision. Under this circumstance, the 3D registration model can be quickly adapted to new point clouds with no or limited labeled training data.

- In this paper, we propose a novel variational encoder that maps shape representations to a distribution over 3D registration tasks, contributing to robust learning of priors over the 3D registration function space.

- In this paper, we observe that different regions of a 3D shape vary in their geometric structures which enables us to propose a region-aware flow regression module instead of the shape-conditioned one.

- In this paper, we compared our 3D Meta-Registration to other state-of-the-art ones on widely used benchmark datasets and demonstrated superior registration performance over both seen and unseen data.

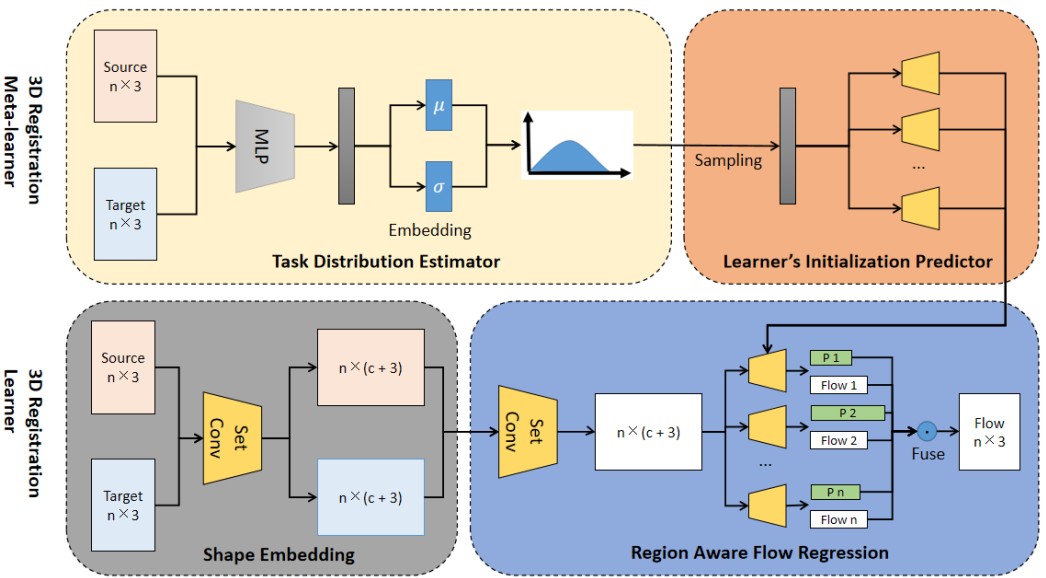

Figure 2: Main Pipeline. Our model includes two networks: 3D registration learner and 3D registration meta-learner. The registration learner includes shape embedding and region aware flow regression. 3D registration meta-learner include task distribution estimator and learner's initialization predictor. Pn denotes the n-th region probability score.

## 2 RELATED WORKS

### 2.1 POINT CLOUD REGISTRATION

In comparison to classical methods Besl & McKay (1992); Yang et al. (2015); Myronenko et al. (2007), learning-based methods have significant advantages in dealing with a large number of datasets by transferring the "knowledge" from registering training pairs to testing pairs. Based on the feature learning structure proposed by PointNet Qi et al. (2017a), Aoki et al. Aoki et al. (2019) proposed PointNetLK for rigid point cloud registration by leveraging Lucas & Kanade algorithm. Based on DGCNN Wang et al. (2019), Wang et al. proposed Deep Closest Point Wang & Solomon (2019a) for learning rigid point cloud registration and PR-Net Wang & Solomon (2019b) for learning partial shapes registration. For non-rigid point cloud registration, Liu et al. proposed FlowNet3D Liu et al. (2019) to learn the points flow field for non-rigid point cloud registration. Wang et al. proposed FlowNet3D++ Wang et al. (2020) on top of FlowNet3D by adding geometric constraints and angular alignment to dramatically improve the alignment performance.

### 2.2 META-LEARNING METHODS

Meta-learning Andrychowicz et al. (2016); Schmidhuber (1992); Hospedales et al. (2020) refers to a subfield of machine learning that learning new concepts and skills much faster and more efficiently given only a small amount of dataset. Parameters prediction Finn et al. (2017; 2018); Lee & Choi (2018) is one of the strategies in meta-learning, which refers to a network trained to predict the parameters of another network so that the first network can encode the related information to the second network which makes the overall network more flexible and adaptive to a particular task. Recently, meta-learning approaches are widely used in computer vision tasks. MANN was proposed by Santoro et al. Santoro et al. (2016) to use an explicit storage buffer which is easier for the network to rapidly incorporate new information and not to forget in the future. Ravi et al. Ravi & Larochelle (2016) use the LSTM to learn an update rule for training a neural network in few-shot learning. Snell et al. Snell et al. (2017) proposed Prototypical networks for few-shot classification task which map the sample data of each class into a metric space by calculating the euclidean distance of prototype representations of each class. In the field of 3D computer vision, Littwin et al. Littwin & Wolf (2019) firstly use a deep neural network to map the input point cloud to the parameters of another

network for the 3D shape representation task. Yang et al. Yang et al. (2020) proposed Meta3D which an external memory is used to store image features and their corresponding volumes. In this paper, we first propose a meta-learning based method with a 3D registration meta-learner, which can learn to learn the registration pattern more efficiently.

## 3 METHODS

We introduce our approach in the following sections. In section 3.1, we state the problem of learning-based registration. Section 3.2 illustrates the 3D registration learner. The 3D registration meta-learner is explained in section 3.3. The definition of the loss function is discussed in section 3.4.

### 3.1 PROBLEM STATEMENT

We need to define the optimization task firstly. For a given training dataset $\mathbf{D} = \{(S_i, G_i)\}$, where $S_i, G_i \subset \mathbb{R}^3$, $S_i$ is the source point cloud and $G_i$ is the target point cloud. We assume the existence of a parametric function $g_\theta(S_i, G_i) = \phi$ using a neural network structure, where $\phi$ is the transformation function (point flow in this paper) which deforms source point cloud towards the target point cloud. We call $g_\theta$ 3D registration learner in this paper and $\theta$ is the set of weights in the 3D registration learner. For previous learning-based network structure, the $\theta$ is optimized using stochastic gradient descent based algorithm for a given dataset:

$$\theta^{\mathbf{optimal}} = \arg\min_\theta[\mathbb{E}_{(S_i, G_i) \sim \mathbf{D}}[\mathcal{L}(S_i, G_i, g_\theta(S_i, G_i))]] \tag{1}$$

where $\mathcal{L}$ represents a similarity measure.

In comparison, we assume that the 3D registration learner $g$ includes two sets of parameters: $\theta_t$ and $\theta_m$. $\theta_t$ is pre-trained from training dataset, but $\theta_m$ is predicted by another parametric function $f_\sigma$ which is called 3D registration meta-learner in this paper. Similarly, we have the desired transformation function $\phi = g_{(\theta_t, \theta_m)}(S_i, G_i)$ and we have $\theta_m = f_\sigma(S_i, G_i)$. For a given training data set, we have:

$$\theta_{\mathbf{t}}^{\mathbf{optimal}}, \sigma^{\mathbf{optimal}} = \arg\min_{\theta_t, \sigma}[\mathbb{E}_{(S_i, G_i) \sim \mathbf{D}}[\mathcal{L}(S_i, G_i, g_{(\theta_t, f_\sigma(S_i, G_i))}(S_i, G_i))]] \tag{2}$$

### 3.2 3D REGISTRATION LEARNER

The 3D registration learner includes two modules: shape embedding (3.2.1) and region aware flow regression (3.2.2). We discuss them in the following subsections.

#### 3.2.1 SHAPE EMBEDDING

For a given pair of input point clouds, we firstly learn to extract the shape features that captures representative and deformation-insensitive geometric information. Let $(S_i, G_i)$ denotes the source and target point clouds and $S_i, G_i \subset \mathbb{R}^3$. $\forall x \in S_i$, we denote the feature of $x$ as $l_x \in \mathbb{R}^c$. Following the recent architecture from PointNet++ Qi et al. (2017b) and FlowNet3D Liu et al. (2019), the first set convolution $g_1 : \mathbb{R}^3 \to \mathbb{R}^c$ is a non-linear MLP-based function $g_1$:

$$l_x = \text{Maxpool}\{g_1(x_j)\}_{||x_j - x|| \leq r \wedge x_j \in S_i} \tag{3}$$

where r is a predefined distance and Maxpool is an element-wise max pooling function.

$\forall x \in S_i$, we further concatenate the coordinates $x$ with the learned feature $l_x$ and we denote it as $[x, l_x] \in \mathbb{R}^{(c+3)}$. Similarly, we can learn the feature of each point in the target point cloud $G_i$. The shape descriptor for source point cloud $S_i$ is: $\{[x, l_x]\}_{x \in S_i}$ and the shape descriptor for target point cloud $G_i$ is: $\{[x, l_x]\}_{x \in G_i}$.

### 3.2.2 REGION AWARE FLOW REGRESSION

Based on the learned shape descriptors for both source point cloud $\{[x, l_x]\}_{x \in S_i}$ and target point cloud $\{[x, l_x]\}_{x \in G_i}$ from previous section, in this section we introduce two more set convolution structures for point flow regression. We define the second set convolution $g_2 : \mathbb{R}^{(2c+9)} \rightarrow \mathbb{R}^{(c+3)}$ to learn the relation information between descriptors of source and target point clouds. $g_2$ is a non-linear MLP-based function. $\forall x \in S_i$, we denote relation tensor $p_x$ as:

$$p_x = \text{Maxpool}\{g_2([x, l_x, y_j, l_{y_j}, x - y_j])\}_{||y_j - x|| \leq r \wedge y_j \in G_i} \tag{4}$$

where [,] denotes concatenation.

Based on the learned relation feature $\{[x, p_x]\}_{x \in S_i}$ from source and target descriptors, we define a set of region aware decoders $\{g_{3,k}\}_{k=1,2,...,K}$, where $K$ is the pre-defined region numbers in dataset **D** and $g_{3,k} : \mathbb{R}^{(c+3)} \rightarrow \mathbb{R}^3$ is a non-learn MLP-based function.

We note the pre-trained weights in $g_3$ as $\theta_t$ and the meta-learned weights in $g_3$ as $\theta_m$. The weights for $g_3$ is the element-wise summation of $\theta_t$ and $\theta_m$. $\forall x \in S_i$, we have the estimated region probability score $p_{x,k}$ and flow $v_{x,k}$ for each region. Notice that we use a softmax function to normalize the output to the estimated region probability score $p_{x,k}$. We have:

$$p_{x,k}, v_{x,k} = g_{3,k}([x, p_x]) \tag{5}$$

Then we use the estimated region probability score $p_{x,k}$ as the weight to further balance among the point flow $v_{x,k}$ for each region. We define the final point flow $v_x$ as:

$$v_x = \sum_{k=1}^{K} p_{x,k} v_{x,k} \tag{6}$$

Therefore, the transformed source shape $S_i' = \{x + v_x\}_{x \in S_i}$.

### 3.3 3D REGISTRATION META-LEARNER

The 3D registration meta-learner includes two parts: task distribution estimator (3.3.1) and learner's initialization predictor (3.3.2). We discuss them in the following subsections.

### 3.3.1 TASK DISTRIBUTION ESTIMATOR

We leverage a variational auto-encoder (VAE) network to learn the task distribution from each pair of input shapes. Instead of learning the detailed features of 3D shapes, we learn the most general global shape information via VAE. More specifically, we use a multi-layer MLP-based function $f_1 : \mathbb{R}^3 \rightarrow \mathbb{R}^v$ to learn the mean and variance of the task distribution for source and target point clouds. We denote the weights in $f_1$ as $\sigma_1$. We denote $\mu$ as the mean of task distribution space, and $\sigma$ as the standard deviation of task distribution space. $\forall (S_i, G_i)$,

$$\mu, \sigma = f_1([\mathbf{x_i}, \mathbf{y_i}])_{\mathbf{x_i} \in \mathbf{S_i}, \mathbf{y_i} \in \mathbf{G_i}} \tag{7}$$

We further sample $L \sim N(\mu, \sigma)$, and $L \subset \mathbb{R}^v$.

### 3.3.2 LEARNER'S INITIALIZATION PREDICTOR

To enable the 3D registration meta-learner to learn the pattern of registration, we introduce a second multi-layer MLP based architecture $f_2 : \mathbb{R}^v \rightarrow \mathbb{R}^w$, where $w$ indicates the dimension of all the weights included in $\theta_2^1, \theta_2^2, \theta_2^3$. We denote the weights in $f_2$ as $\sigma_2$. The meta-learned weights in the 3D registration learner are predicted from the following. $\forall (S_i, G_i)$, we have:

$$\theta_{\mathbf{1}}, ..., \theta_{\mathbf{k}} = f_2([L]) \tag{8}$$

Therefore, for any given pair of source and target point clouds, our 3D registration learner can be accordingly adjusted by adding the meta-learned weights of $\theta_m = \{\theta_1, ..., \theta_k\}$ from the 3D registration meta-learner together with the pre-trained weights of $\theta_l$.

## 3.4 LOSS FUNCTION

For the target point cloud $G_i$ and transformed source point cloud $S'_i$, we define the loss function in this section. Assuming that we have ground truth flow field $\{v^*_x\}_{x \in S_i}$, we use the simple $L_1$ loss between predicted flow field and ground truth flow field with a cycle consistency term as regularization. We note $v'_x$ as the predicted flow from $S'_i$ to $S$. The loss function is defined as:

$$\mathcal{L}(S'_i, G_i) = \frac{1}{|S'_i|} \sum_{x \in S_i} \{||v_x - v^*_x|| + \lambda_1 ||v_x + v'_x||\} + \lambda_2 KL(N(\mu, \sigma), N(0, 1)) \quad (9)$$

, where $| * |$ denotes the set cardinality and $\lambda_1$ is a pre-defined hyper-parameter to balance the two terms. $\lambda_2$ is the balance term for the KL divergence regularization

## 4 EXPERIMENT

In this section, we describe the dataset and experimental settings in section 4.1 and section 4.2. In section 4.3 we perform an ablation study on FlyingThings3D dataset to demonstrate our model's performance. In section 4.4, we conduct a series of experiments to verify the effect of our model in the registration of unseen data set from different categories on ModelNet40 dataset. We demonstrate the effectiveness of our proposed method on a large-scale synthetic dataset (FlyingThings3D) in section 4.5, and in section 4.6 we show our model's generalization capacity on registration of real Lidar scans from KITTI.

### 4.1 DATASET PREPARATION

**ModelNet40:** This dataset contains 12311 pre-processed CAD models from 40 categories. For each 3D point object, we uniformly sample 2048 points from its surface. For each source shape $S_i$ we generate the transformed shapes $G_i$ by applying a rigid transformation defined by the rotation matrix which is characterized by 3 rotation angles along the x-y-z-axis, where each value is uniformly sampled from $[0, 45]$ unit degree, and the translation which is uniformly sampled from $[-0.5, 0.5]$. For the testing dataset of ModelNet40 with noise, we prepare three types of noise added on both source and target point set. To prepare the position drift (P.D.) noise, we applied a zero-mean and 0.05 standard deviation Gaussian to each sample from the point set. To prepare the data incompleteness (D.I.) noise, we randomly choose a point and keep its nearest 1536 points from both the source and target point set. To prepare the data outlier (D.O.) noise, we randomly add 148 points as outliers, sampled from a $[-10, 10] \times [-10, 10] \times [-10, 10]$ uniform distribution.

**FlyingThings3D:** The FlyingThings3D dataset is an open-source collection, which consists of more than 39000 stereo RGB images with disparity and optical flow ground truth. By following the process procedures provided by FlowNet3D, we generate 3D point clouds and registration ground truth using the disparity map and optical map rather than using RGB images.

**KITTI:** Another dataset used in this paper is the KITTI scene flow dataset, which consists of 200 training scenes and 200 test scenes. Following previous work FlowNet3D, we use the pre-processed point clouds data which is generated using the original disparity map and ground truth flow.

### 4.2 SETTINGS

In our model, the batch size is set to 16 and we use Adam optimizer as our optimizer. For the 3D registration learner, the first set convolution includes 6 MLPs with dimensions of (32, 32, 64, 64, 64, 128). The second set convolution includes 6 MLPs with dimensions of (128, 128, 256, 256, 256, 512). The third set convolution includes 6 MLPs with dimensions of (256,256,128,128,256,256) and region-aware decoders includes 8 fully connected layers with dimensions of (128,3). For the 3D registration meta-learner, we use 3 MLPs with dimensions of (32, 64, 128) with a fully connected layer with dimension (256) for learning the mean of VAE and one fully connected layer with dimension

Table 1: Ablation study. End-point-error (EPE) and estimation accuracy(ACC) with 0.05 and 0.1 threshold are used for evaluation of point cloud registration performance to compare our model with different settings on the FlyingThings3D dataset.

| Models | EPE | ACC(0.05) | ACC(0.1) |
|---|---|---|---|
| Learner with Weight-Setting-A | 0.1694 | 25.37% | 57.85% |
| Learner with Weight-Setting-B | 0.1503 | 27.88% | 60.22% |
| Learner with Weight-Setting-C | 0.1453 | 28.67% | 61.65% |
| Learner with Weight-Setting-D | **0.1437** | **29.92**% | **62.21**% |

Table 2: Results on the 20 unseen testing categories of the ModelNet40 dataset with noise.

| Method | Noise | EPE | ACC (0.05) | ACC (0.1) |
|---|---|---|---|---|
| w/o meta-learner | - | 0.3648 | 10.01% | 45.18% |
| w/ meta-learner | - | **0.2342** | **34.01**% | **75.56**% |
| w/o meta-learner | P.D. | 0.4392 | 6.75% | 34.47% |
| w/ meta-learner | P.D. | **0.3128** | **20.92**% | **61.30**% |
| w/o meta-learner | D.I. | 0.3749 | 11.67% | 46.07% |
| w/ meta-learner | D.I. | **0.2689** | **24.04**% | **65.84**% |
| w/o meta-learner | D.O. | 0.5738 | 4.81% | 24.05% |
| w/ meta-learner | D.O. | **0.4156** | **10.49**% | **40.91**% |

(256) for learning the standard deviation of VAE. Then we have three fully connected layers with dimensions of 128, 512, 2048, and the number of all parameters in the set convolution layers. We use the ReLU activation function and implement batch normalization for every MLP layer except the last output layer. We set the learning rate as 0.001 with exponential decay of 0.7 at every 20000 steps. For the balance terms in loss function, $\lambda_1$ is set to 0.3 and $\lambda_2$ is set to 1 in this paper.

For evaluation of point cloud registration performance, we use 3D endpoint error (EPE) and point cloud registration estimation accuracy (ACC) with different thresholds as introduced in FlowNet3D. The 3D EPE measures the L2 distance between the point cloud registration estimation and ground truth. A lower EPE value indicates a better estimation of the point flow field. ACC measures the portion of the points with estimated flow field error which is less than one selected threshold among all the points.

## 4.3 ABLATION STUDY

**Experiment setting.** For this experiment, we use the FlyingThings3D dataset for demonstration. We test four settings of our model for comparison of different weight generation methods. In the weight-setting-A, without our meta-learning strategy, the network parameters of the 3D registration learner are optimized from the training dataset. In the weight-setting-B, the network parameters of the 3D registration learner are meta-learned by the 3D registration meta-learner. Note that in the weight-setting-B, we directly use MLPs to learn the function prior which is further send to the prediction of the learner as input to predict the parameters of the 3D registration Learner. In the weight-setting-C, instead of directly using the MLPs to map the input shape to the function prior, we add a variational auto-encoder(VAE) to learn the task distribution of the input 3D shape, which can be further sampled as the function prior over task distribution for the learner predictor to estimate the optimal weights of the 3D registration learner. In the weight-setting-D, we use the region aware decoder to estimate the probability score and point flow for each region which is further used to weighted fuse the multiple transformations to get the final point flow. The quantitative results are demonstrated in Table 1.

**Results.** From the first two rows, we notice that the results of weight-setting-B with our meta learning strategy are better than the results of weight-setting-A with all weights predicted from the registration model learning module. This indicates that our meta-learning method can learn the prior over 3D registration function space and benefit the 3D registration learner from good meta-learned initialized weights. Moreover, as indicated in row 2 and 3, the experimental results of the weight-setting-C are better than the weight-setting-B, which indicates that the task distribution generated by the VAE in the 3D registration meta-learner can gather the prior over the 3D registration function

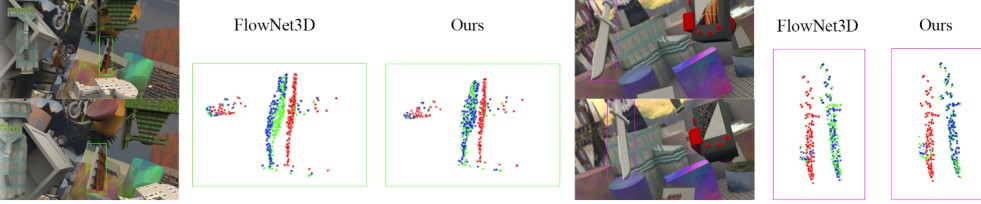

Figure 3: Selected point cloud registration qualitative results on the FlyingThings3D dataset. Red points represent the source point cloud. Green points represent the target point cloud. Blue points represent the transformed source point cloud.

space and further sampled for the learner predictor to predict the optimal parameters of the 3D registration learner. The weight-setting-D achieves 0.1437 EPE which indicates that the proposed region aware decoder with weighted fusion can further improve the performance. In the rest of the experiments, we use this setting as our default model.

### 4.4 EXPERIMENTS ON UNSEEN CATEGORY DATASET

**Experiment setting.** We use the first 20 categories in the ModelNet40 dataset as our training dataset. We use the last 20 categories in ModelNet40 as our testing dataset. We compare our model with meta-learner and without meta-learner. Specifically, we use the last 20 categories with different noise in ModelNet40 as our testing dataset. We do the exact fair comparison to show the effect of our proposed meta-learner in dealing with the unseen dataset. We list the quantitative results in Table 2.

Table 3: Results on the Flx yingThings3D test dataset.

| Method | EPE | ACC (0.05) | ACC (0.1) |
|---|---|---|---|
| ICP | 0.5019 | 7.62% | 7.62% |
| FlowNet3D | 0.1694 | 25.37% | 57.85% |
| FlowNet3D++ | 0.1553 | 28.50% | 60.39% |
| MeteorNet | 0.209 | – | 52.12% |
| HPLFlowNet | 0.1453 | 29.46% | 61.91% |
| Ours | **0.1437** | **29.92**% | **62.21**% |

**Results.** As shown in Table 2, for all these three types of noise (P.D, D.I, and D.O noise), with the meta-learner our model achieves much lower EPE and higher accuracy compared to the model without meta-learner on the unseen 20 testing categories. This result indicates that our model with meta-learner can be less affected by all these different noise patterns in comparison with the model without meta-learner.

### 4.5 EXPERIMENTS ON FLYINGTHINGS3D DATASET

**Experiment setting:** From nearly 39000 models in the FlyingThings3D dataset, we randomly select 20000 models for training and 2000 models for testing. We sample 2048 points from each point cloud. To show the efficiency of our proposed model, we compare it with ICP(iterative closest point) and three settings of FlowNet3D. FlowNet3D refers to the optimal model in Liu et al. (2019). We also compare

Table 4: Results on the KITTI dataset.

| Method | EPE | ACC (0.05) | ACC (0.1) |
|---|---|---|---|
| FlowNet3D | 0.2113 | 8.79% | 32.85% |
| SPLATFlowNet | 0.1988 | 21.74% | 53.91% |
| MeteorNet | 0.2510 | - | - |
| FlowNet3D++ | 0.2530 | - | - |
| Ours | **0.1456** | **24.63**% | **55.44**% |
| Ours(self-supervised) | **0.1196** | **29.49**% | **61.19**% |

our method with state-of-the-art methods FlowNet3D++ Wang et al. (2020) and HPLFlowNet Gu et al. (2019). We list the quantitative results in Table 3 and we demonstrate selected qualitative results for in Figure 3.

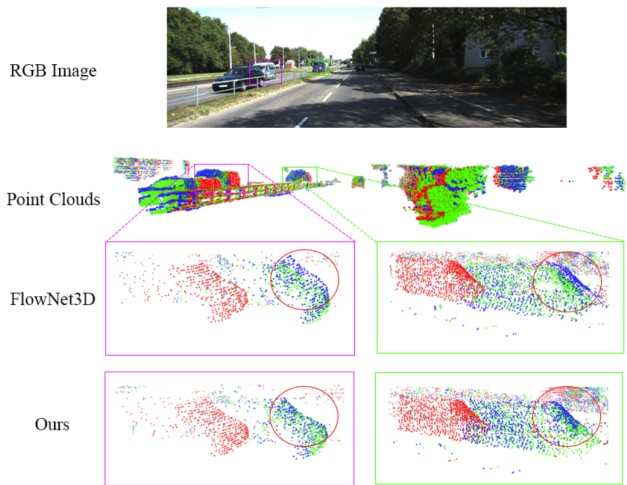

Figure 4: Qualitative results of point cloud registration on the KITTI dataset. Red points represent the source point cloud. Green points represent the target point cloud. Blue points represent the transformed source point cloud.

**Results.** As shown in Table 3, our method achieves lower EPE (0.1437) compared to FlowNet3D (0.1694) and ICP (0.5019). As to the registration estimation accuracy (ACC), our method achieves significantly better results with 29.92% for the threshold 0.05 and 62.21% for the threshold 0.1, which is better than 25.37% for the threshold 0.05 and 57.85% for the threshold 0.1 achieved by FlowNet3D. Moreover, our proposed method outperforms the state-of-the-art methods FlowNet3D++ and HPLFlowNet. In addition, from the qualitative results shown in Figure 3, we notice that the alignment of source and target point clouds is more accurate for our method. For example, we can clearly see the gap between green (target) and blue (transformed source) points for the left case in the result of FlowNet3D.

### 4.6 EXPERIMENTS ON KITTI DATASET

**Experiment setting.** The KITTI scene flow dataset includes 200 stereo images. Following the evaluation standards claimed in FlowNet3D for a fair comparison, we use the first 150 images from the KITTI scene flow dataset as the testing dataset to evaluate our model. Note that our model is only trained on the training dataset of FlyingThings3D.

**Results.** The results shown in Table 4 demonstrates that our model trained on FlyingThings3D achieves lower EPE and better ACC on KITTI in comparison to the results achieved by FlowNet3D, SPLATFlowNet Balakrishnan et al. (2018). For the qualitative results shown in Figure 4, we can clearly see that our registration result for the two cars on the left side is better since all the blue points are almost overlapped with the green points. In comparison, the result of FlowNet3D shows a gap between blue and green points.

## 5 CONCLUSION

This paper introduces a novel meta-learning-based approach to our research community for point cloud registrations. In contrast to recent proposed learning-based methods, our method leverages a 3D registration meta-learner to learn the prior over 3D registration function space, the 3D registration meta-learner can accordingly predict an optimal structure for the 3D registration learner to address the desired transformation for registration. To the best of our knowledge, our method firstly leveraged a meta-learning based structure for this task and we achieved superior point cloud registration results on the unseen dataset.

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
