# OpenReview forum: "3D Meta-Registration: Meta-learning 3D Point Cloud Registration Functions"
_ICLR.cc/2022/Conference — ICLR 2022 Submitted_

### Official Review · Reviewer_t8Fa · 2021-11-02

**Correctness:** 2
**Technical Novelty And Significance:** 2
**Empirical Novelty And Significance:** 2
**Recommendation:** 5
**Confidence:** 4

**Main Review:**

Strengths:
The idea looks new to me. The results are promising but are still not solid to support the claim (well addressing the generalization problem).

Weakness:
Though this work is about 3D point cloud registration, many important works about 3D point cloud registration are missing. For example, feature-based methods like FCGF, D3Feat, and end-to-end methods such as Global Point Registration, PREDATOR, PR-Net, RPM-net，Deep closet point, etc.

The proposed method does not show its superior performance on generalization in the experiments. The only experiment for evaluation of generalization capability is the KITTI test, where the model was trained on flyingthings3d and tested on KITTI.  I think more tests should be conducted on more datasets and compared with some state-of-the-art methods such as PREDATOR and RPM-Net.

**Summary Of The Paper:**

This paper presents a new architecture for point cloud registration. Different from existing methods, the proposed architecture consists of two stages. The first one is called meta learner, which is used to predict a task distribution and sample the key parameters for the second stage - 3D registration learner.
By separating the network into two different parts, the learned model may be better at generalization.

Experiments were conducted on ModelNet40, FlyingThings3D, and KITTI datasets. The results show the proposed method outperforms some existing methods, including FlowNet3D, FlowNet3D++, and HPLFlowNet.

**Summary Of The Review:**

1. The proposed idea is new.
2. Many related works are missing.
3. Experiments are not solid to support the idea.

---

### Official Review · Reviewer_xfRw · 2021-11-03

**Correctness:** 3
**Technical Novelty And Significance:** 3
**Empirical Novelty And Significance:** 3
**Recommendation:** 3
**Confidence:** 4

**Main Review:**

The objective of the work: This paper proposes a learning method to improve the generalization ability for the point cloud registration tasks.

Strong points: This overall idea is a hot topic in the point cloud area. The concept of this meta-learning is interesting.

Weak points: (1). The paper claimed to improve the generalization ability, while the experiments are extremely lacked to support this claim. All the other recent point cloud registration methods (e.g, FMR[1], DeepGMR[2], RGM[3], PntLK++[4]) achieve very well in this generalization setting. (2) The paper claimed to solve the point cloud registration problem. However, the experiments have not been compared with the recent state-of-the-art point cloud registration methods, such as FMR, DeepGMR, RGM, PntLK++. The current experimental results in Table 2 seems worse than the state-of-the-art registration methods.

[1]. Huang, Xiaoshui, Guofeng Mei, and Jian Zhang. "Feature-metric registration: A fast semi-supervised approach for robust point cloud registration without correspondences." Proceedings of the IEEE/CVF Conference on Computer Vision and Pattern Recognition. 2020.
[2]. Yuan, Wentao, et al. "Deepgmr: Learning latent gaussian mixture models for registration." European Conference on Computer Vision. Springer, Cham, 2020.
[3]. Fu, Kexue, et al. "Robust Point Cloud Registration Framework Based on Deep Graph Matching." Proceedings of the IEEE/CVF Conference on Computer Vision and Pattern Recognition. 2021.
[4]. Li, Xueqian, Jhony Kaesemodel Pontes, and Simon Lucey. "PointNetLK Revisited." Proceedings of the IEEE/CVF Conference on Computer Vision and Pattern Recognition. 2021.


**Summary Of The Paper:**

This paper proposes a learning method to estimate the transformation matrix directly using neural networks. This paper tried to leverage the meta-learning strategy to improve the generalization ability.

**Summary Of The Review:**

The overall idea is interesting. However, the experiments are much lacking behind to support the claims, and the current experiments have not compared with the state-of-the-art registration methods. I think this work is ongoing work and need to do more investigation.

---

### Official Review · Reviewer_woQH · 2021-11-05

**Correctness:** 2
**Technical Novelty And Significance:** 3
**Empirical Novelty And Significance:** 2
**Recommendation:** 3
**Confidence:** 4

**Main Review:**

The design of this proposed network is fairly interesting, but I am not convinced that all claims are properly substantiated. Additionally, the paper itself isn't very polished and leaves me with several concerns regarding its effectiveness for the proposed applications.

First, 3d point cloud registration typically refers to finding a single rigid transformation in SE(3) that best aligns two point sets. The title mentions point cloud registration, but the method itself and the methods it is compared against are scene flow methods. Scene flow differs from 3d point cloud registration in that each point is assigned a different transformation in SE(3), typically in a 1-to-1 manner as a 3D analogue of optical flow. Given the non-standard use of technical terms, I question how this method is to be used. If the main comparative work is FlowNet3D and FlowNet3D++, it would seem that this paper should be more aligned with other scene flow works than registration (e.g. you also compare against ICP, which *is* a 3d point cloud registration algorithm).

The qualitative results are not all that convincing to me, and I don't know whether or not the result was cherry-picked or demonstrates the typical performance difference. Much of the details I would need to reimplement this method (including explanations of metric and experimental procedure) just refer to FlowNet3D. I think to make the paper more self-contained, it would be nice to include a little more detail instead of requiring reading FlowNet3D to understand your evaluation.

Training on FlyThings3D and testing on KITTI, showing good results, is encouraging to me that your method is actually achieving some superior generalizability performance with its meta-learning design. However, I am not quite sure if your ablations prove your claims since one could attribute the superior performance to just having a higher capacity network when the meta-learner is enabled? Also, why do you elementwise add the estimated meta parameters to the learned parameters? It would nice to see an ablation where you either fully predict the parameters (a traditional hypernetwork approach), or concatenate instead of adding, or use the predicted parameters just for layer normalization (which is more commonly seen in approaches like conditional batchnorm, or FiLM, or even in things like AdaIn or StyleGAN).

The text itself needs some work-- the citations seem to have a latex problem and break up the text, disrupting the flow and making everything hard to parse. Some of the math exposition is a little imprecise. For example, in Equation 1, shouldn't it be argmax if \mathcal{L} is a similarity measure? Also, S_i and G_i are *not* subsets of R^3. A subset of R^3 is a single 3D point. S_i and G_i, since they are 3d point clouds, are a set of points in R^3. Typically, I would expect to see something like S_i \in R^{N \times 3}.

Other small things: uniformly sampling rotation angles on each three rotation axes independently results in a biased sampling of SO(3). Axis-angle representation is better suited for unbiases SO(3) sampling of rotation angles under 45 degrees. In Table-1, you should have understandable shorthand instead of A,B,C,D-- I find this really hard to cross-reference.

**Summary Of The Paper:**

A scene flow estimation network is proposed with a meta-learning hypernetwork type design: the top level estimates the parameters of the bottom level network, which combines with meta-learned parameters to enable better generalization properties.

**Summary Of The Review:**

Overall, I think this paper just has too many problems for me to recommend acceptance. I'm not fully convinced by the approach to meta-learn in function space-- I did not see a convincing argument explain how exactly this is necessary (e.g. a toy experiment to demonstrate its necessity would be nice)-- and the ablation results seem to be explainable by just having a higher capacity and therefore higher performing network. Lastly, I am a little confused whether this is a scene flow or a point cloud registration paper-- it would appear to be the former though the title and intro seem to indicate the former.

---

### Official Review · Reviewer_eJC8 · 2021-11-06

**Correctness:** 4
**Technical Novelty And Significance:** 2
**Empirical Novelty And Significance:** 2
**Recommendation:** 3
**Confidence:** 4

**Main Review:**

Presentation is in general deficient; often the text seems vague. I will give some examples next. I believe that concise explanations and providing examples could help in communicating ideas in this paper.

I feel the abstract could be improved to be more precise. For example, when saying "we define each task", results unclear what are those tasks or why is it not a single task.

The following affirmation seems vague to me "In comparison to iterative registration methods, learning-based methods have advantages in dealing with a large number of datasets since learning-based methods can transfer the registration pattern from one dataset to another one." I feel we can also transfer patterns (parameters?) from different datasets when using traditional methods.

I am not sure about the meaning of optimal in the following sentence: "The meta-learner is responsible for providing the optimal initialization of a 3D registration learner". My intuition tells me that an optimal initialisation does not require any adjustment; hence no need for a 3D registration learner.

I am not sure what the text is trying to comunicate in the following sentence: "... our meta-learning-based approach gains competitive advantages over existing generalization methods that our method can uniquely parameterize the 3D registration function for each pair of shapes to provide 3D point cloud registration."



On experimentation over the KITTI dataset:
----------------------------------------------------------
I think the text should explain the point cloud pair generation (source, target) from the stereo vision information. How many pairs were considered?

I recommend report also quantitative results for the KITTI dataset.


Other comments
-----------------------
I suggest rewriting section 2.2 (META-LEARNING METHODS) to contrast the proposed approach against existent ones. My impression of the review here is a list of related approaches but without comparing them with the propositions here. Beyond the particular problem here (3D registration), what do other methods have in common with the proposed one? What is new?

What is the notation "~" in Eq. (1)?

Why theta_m is not in Eq. (2)?

What is the difference between theta_m and sigma?

What is the self-supervised variant in Table 4?


**Summary Of The Paper:**

This paper addresses point cloud registration from a meta-learning perspective to quickly adapt with limited training data. The main idea is using a meta-learner is to initialise a 3D registration learner. The meta-learner predicts a prior registration that can rapidly adapt to new registration problems. Experimental results on several datasets (ModelNet, FlyingThings3D, and KITTI) showed superior performance over FlowNet3D.


**Summary Of The Review:**

I found the presentation deficient and my impression is that despite the results seeming promising, this paper requires substantial improvements to be ready for another review process.

This paper seems correct in methodology however I did not check everything, in part of difficulties related to the presentation.

---

### Decision · Program_Chairs · 2022-01-20

**Decision:**

Reject

**Comment:**

This paper applies a metalearning strategy to point cloud registration, which refines 3D registration networks to improve performance on specific datasets/settings.  Reviews for this paper recognized its potential interest but uniformly highlighted that the work is lacking in polish---both from an expository perspective and in terms of experiments.  Questions included whether the experiments truly support the claim of generalization, and whether the work would be better considered as a method for scene flow.  Authors did not rebut these points, so I am recommending rejection.